# Perioperative PD-1/PD-L1 inhibitors for resectable non-small cell lung cancer: A meta-analysis based on randomized controlled trials

Hai Huang, Lianyun Li, Ling Tong, Houfu Luo, Huijing Luo, Qimin Zhang ⓘ *

Department of Oncology, Taihe People's Hospital, Taihe, China

* thxrmyyzlk@163.com

## Abstract

### Background

PD-1/PD-L1 inhibitors (PI) have shown promising results in both neoadjuvant and adjuvant therapies for resectable non-small cell lung cancer (NSCLC). However, substantial evidence from large-scale studies is still lacking for their use in the perioperative setting (neoadjuvant plus adjuvant). This meta-analysis aims to evaluate the integration of perioperative PI (PPI) with neoadjuvant chemotherapy for resectable NSCLC.

### Methods

To identify appropriate randomized controlled trials (RCTs), we thoroughly explored six different databases. The primary endpoint was survival, while the secondary measures included pathological responses and adverse events (AEs).

### Results

Six RCTs involving 2941 patients were included. The PPI group significantly improved overall survival (OS) (hazard ratio [HR]: 0.62 [0.51, 0.77]), event-free survival (EFS) (HR: 0.57 [0.51, 0.64]), pathological complete response (risk ratio [RR]: 5.81 [4.47, 7.57]), and major pathological response (RR: 2.60 [1.77, 3.82]). Benefits in EFS were seen across all subgroups. OS rates at 12–48 months and EFS rates at 6–48 months were higher in the PPI cohort. Furthermore, the advantages in OS and EFS increased with prolonged survival times. The PPI group also exhibited higher rates of surgery and R0 resections. However, the PPI group experienced more grade 3–5 AEs, serious AEs, and treatment discontinuations due to AEs.

### Conclusions

The integration of perioperative PI with neoadjuvant chemotherapy can significantly improve survival and pathological responses for resectable NSCLC. However, the increased incidence of grade 3–5 AEs must be carefully evaluated.

**Data Availability Statement:** All relevant data are within the paper and its Supporting Information files.

**Funding:** The author(s) received no specific funding for this work.

**Competing interests:** The authors have declared that no competing interests exist.

**Abbreviations:** AE, Adverse event; ALT, Alanine aminotransferase; AST, Aspartate aminotransferase; CI, Confidence interval; CR, Complete response; DCR, Disease control rate; ECOG PS, Eastern Cooperative Oncology Group Performance Status; EFS, Event-free survival; EFSR, Event-free survival rate; GRADE, Grading of Recommendations, Assessment, Development, and Evaluation; HR, Hazard ratio; irAEs, Immune-related adverse events; M/F, Male/Female; MPR, Major pathologic response; PCR, Pathological complete response; ORR, Objective response rate; OS, Overall survival; OSR, Overall survival rate; PD-1, Programmed cell death protein 1; PD-L1, Programmed cell death 1 ligand 1; PICOS, Participants, Intervention, Control, Outcomes, Study design; PI, PD-1/PD-L1 inhibitors; PPI, Perioperative PD-1/PD-L1 inhibitors; PR, Partial response; PRISMA, Preferred Reporting Items for Systematic Reviews and Meta-Analyses; RCT, Randomized controlled trial; RR, Risk ratio; SCC, Squamous cell carcinoma; SD, Stable disease; TNM, Tumor Node Metastasis; TPS, Tumor cell proportion score; TRAEs, Treatment-related adverse events.

## Introduction

Lung cancer remains a major cause of cancer-related deaths globally, with non-small cell lung cancer (NSCLC) accounting for approximately 85% of all cases [1]. Among NSCLC patients, those with resectable disease present a unique opportunity for curative surgical intervention. However, the high recurrence rates following surgery underscore the need for effective perioperative treatment strategies [2]. Recently, PD-1/PD-L1 inhibitors (PI) have emerged as promising therapeutic options, transforming the treatment landscape for NSCLC [3]. Various studies have confirmed the efficacy of PI in both neoadjuvant and adjuvant settings [4, 5]. For instance, neoadjuvant therapy with PI has demonstrated a reduction in tumor burden and improved surgical outcomes by enhancing pathological responses [4]. Similarly, adjuvant therapy with these inhibitors has shown improved survival rates by targeting residual disease and preventing recurrence [5]. Pasqualotto et al.'s meta-analysis, based on seven RCTs, also confirmed the role of PI in both standalone neoadjuvant and adjuvant therapy for resectable NSCLC [6]. Despite these successes, significant controversies and gaps in evidence remain, particularly regarding the integration of these treatments into a comprehensive perioperative approach.

One of the primary controversies in this field involves the optimal timing and sequencing of PI in conjunction with chemotherapy [7]. While neoadjuvant chemotherapy has long been a standard to downstage tumors and eradicate micrometastases, the addition of PI in both the neoadjuvant and adjuvant settings (perioperative PI [PPI]) has not been thoroughly investigated in large sample meta-analyses. The hypothesis that combining chemotherapy with PI can elicit a stronger anti-tumor immune response is compelling [8–13]. Chemotherapy can cause immunogenic cell death, which potentially enhances the efficacy of PI by increasing tumor antigen presentation and T-cell infiltration. However, this theoretical synergy requires validation through rigorous clinical trials [14].

By pooling data from high-quality randomized controlled trials (RCTs), our analysis aims to provide a comprehensive assessment of the impact of PPI on key outcomes, including overall survival (OS), event-free survival (EFS), pathological response, and adverse events (AEs).

## Materials and methods

### Search strategy

MeSH terms such as "PD-1/PD-L1 (see S1 Table for details)", "Chemotherapy", "Lung cancer", and "Randomized" were utilized. We thoroughly searched six databases, including PubMed, ScienceDirect, the Cochrane Library, Scopus, EMBASE, and Web of Science. The search period covered from inception to June 15, 2024 (**S1 Table**). Furthermore, the reference lists of the selected studies were scrutinized to identify additional eligible RCTs.

### Selection criteria

Inclusion criteria (PICOS):

1. Participants (P): resectable NSCLC.

2. Intervention (I) and control (C): PPI group (PPI plus neoadjuvant chemotherapy) versus Chemotherapy group (neoadjuvant chemotherapy).

3. Outcomes (O): survival, surgery condition, pathological response, and safety.

4. Study design (S): RCTs.

Exclusion criteria: animal experiments, reviews, meta-analyses, case reports, and studies missing key data.

## Data extraction

Data included study characteristics (phase, period, etc.), patient demographics (sex, histologic classification, etc.), survival metrics (OS and EFS), survival rates (OS rate [OSR] and EFS rate [EFSR]), pathological responses (objective response rate [ORR], major pathologic response [MPR], etc.), and AEs (total, grade 3–5, etc.). Data were independently extracted by two researchers, and discrepancies were resolved through re-evaluation (**S1 File**).

## Outcome assessments

The OSR and EFSR were analyzed at 6–48 months. Subgroup analyses of EFS were conducted according to age, sex, smoking status, Eastern Cooperative Oncology Group Performance Status (ECOG PS), race, geographic region, pathological stage, histologic classification, and PD-L1 TPS.

## Quality assessment

We employed the Cochrane Risk Assessment Tool and the Jadad scale to evaluate the quality of RCTs, with the latter rating studies up to 5 points based on randomization, blinding, and participant inclusion, considering scores of 3 or more as high quality [15, 16]. The GRADE approach was used to assess the reliability of the results [17].

## Statistical analysis

Review Manager 5.3, Stata 12.0 and SPSS 15.0 were used for data analysis. We employed hazard ratios (HR) for evaluating survival outcomes and risk ratios (RR) for dichotomous outcomes. Heterogeneity was assessed using the $I^2$ statistic and the $\chi^2$ test. A fixed-effects model was selected when $I^2$ was less than 50% or the P-value was above 0.1, indicating low heterogeneity; otherwise, a random-effects model was used. Funnel plots were used to assess publication bias. Statistical significance was defined as P < 0.05. (PROSPERO ID: CRD42024563648).

# Results

## Search results

Our meta-analysis incorporated RCTs involving 2941 patients: AEGEAN, CheckMate 77T, KEYNOTE-671, NADIM II, Neotorch, and RATIONALE-315 (**Fig 1**) [8–13]. **Table 1** presents an overview of the baseline characteristics of these studies. Among these, three were global multicenter trials [8–10], two were conducted in China [12, 13], and one was based in Spain [11]. According to **S1 Fig** and **S2 Table**, all studies exhibited high quality. The GRADE approach was utilized to assess the quality of the results, which ranged from medium to high (**S3 Table**).

## Survival

In the PPI group, OS improved significantly, with an HR of 0.62 [0.51, 0.77] (**Fig 2**). Additionally, the OSR at 12 to 48 months was higher in this group (**S2 Fig**). The benefits in OSR became more evident as the survival time lengthened (**Fig 3A and 3C**).

In the PPI group, EFS also improved significantly, with an HR of 0.57 [0.51, 0.64] (**Fig 2**). The EFSR at 6 to 48 months was higher in the PPI group (**S3 Fig**). The benefits in EFSR

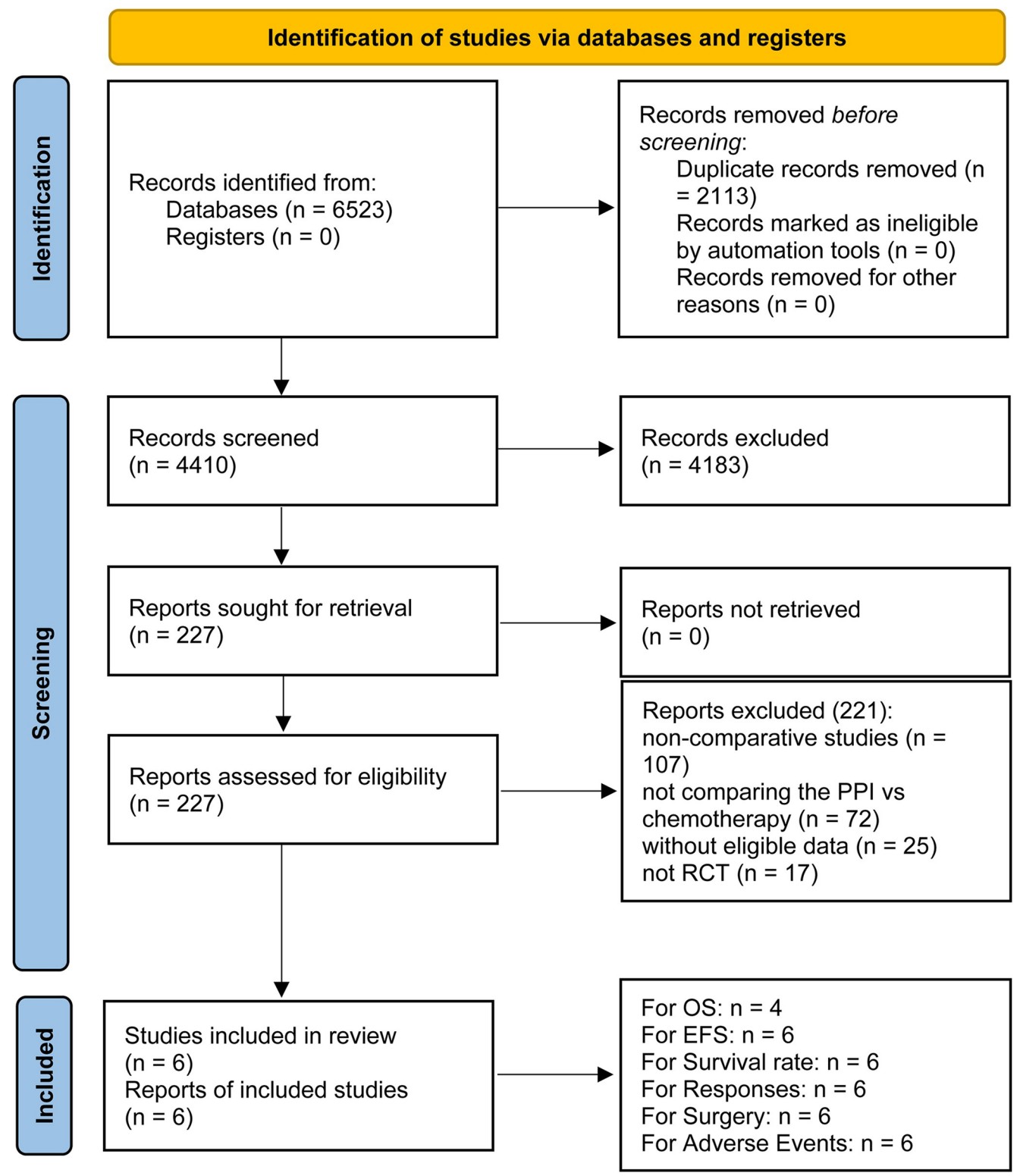

**Fig 1. Flow chart.**

Table 1. Baseline characteristics of the included studies.

| Study | Phase | Period | Country | Groups | Patients | Sex (M/F) | Age (Mean, year) | ECOG PS | | Histologic classification | | TNM Stage | | | PD-1/PD-L1 type | Follow up (months) |
|---|---|---|---|---|---|---|---|---|---|---|---|---|---|---|---|---|
| | | | | | | | | 0 | 1 | SCC | Non-SCC | II | IIIA | IIIB | | |
| AEGEAN (NCT03800134) | III | 2019.01–2022.04 | Global multicenter | PPI | 366 | 252/114 | 65 | 251 | 115 | 169 | 196 | 104 | 173 | 88 | Durvalumab | 34.0 |
| Heymach 2023 [8] | | | | Chemotherapy | 374 | 278/96 | 65 | 255 | 119 | 191 | 179 | 110 | 165 | 98 | | |
| CheckMate 77T (NCT04025879) | III | 2019.11–2022.04 | Global multicenter | PPI | 229 | 167/62 | 66 | 147 | 82 | 116 | 113 | 81 | 146 | | Nivolumab | 25.4 |
| Cascone 2024 [9] | | | | Chemotherapy | 232 | 160/72 | 66 | 141 | 91 | 118 | 114 | 81 | 149 | | | |
| KEYNOTE-671 (NCT03425643) | III | 2018.04–2021.12 | Global multicenter | PPI | 397 | 279/118 | 63 | 253 | 144 | 226 | 171 | 118 | 217 | 62 | Pembrolizumab | 25.2 |
| Wakelee 2023 [10] | | | | Chemotherapy | 400 | 284/116 | 64 | 246 | 154 | 173 | 227 | 121 | 225 | 54 | | |
| NADIM II (NCT03838159) | II | 2019.06–2021.02 | Spain | PPI | 57 | 36/21 | 65 | 31 | 26 | 21 | 36 | 0 | 44 | 13 | Nivolumab | 26.1 |
| Provencio 2023 [11] | | | | Chemotherapy | 29 | 16/13 | 63 | 16 | 13 | 14 | 15 | 0 | 24 | 5 | | |
| Neotorch (NCT04158440) | III | 2020.03–2023.06 | China | PPI | 202 | 181/21 | 62 | 70 | 132 | 157 | 45 | 0 | 136 | 66 | Toripalimab | 18.3 |
| Lu 2024 [12] | | | | Chemotherapy | 202 | 189/13 | 61 | 73 | 129 | 157 | 45 | 0 | 137 | 65 | | |
| RATIONALE-315 (NCT04379635) | III | 2020.05-2023.08 | China | PPI | 226 | 205/21 | 62 | 143 | 83 | 179 | 45 | 93 | 133 | 0 | Tislelizumab | 22.0 |
| Zhang 2023 [13] | | | | Chemotherapy | 227 | 205/22 | 63 | 154 | 73 | 175 | 50 | 92 | 135 | 0 | | |

**Abbreviations:** AE: Adverse event; ECOG PS: Eastern Cooperative Oncology Group Performance Status; PD-1: Programmed cell death protein 1; PD-L1: Programmed cell death 1 ligand 1; PPI: Perioperative PD-1/PD-L1 inhibitors; SCC: Squamous cell carcinoma; TNM: Tumor Node Metastasis.

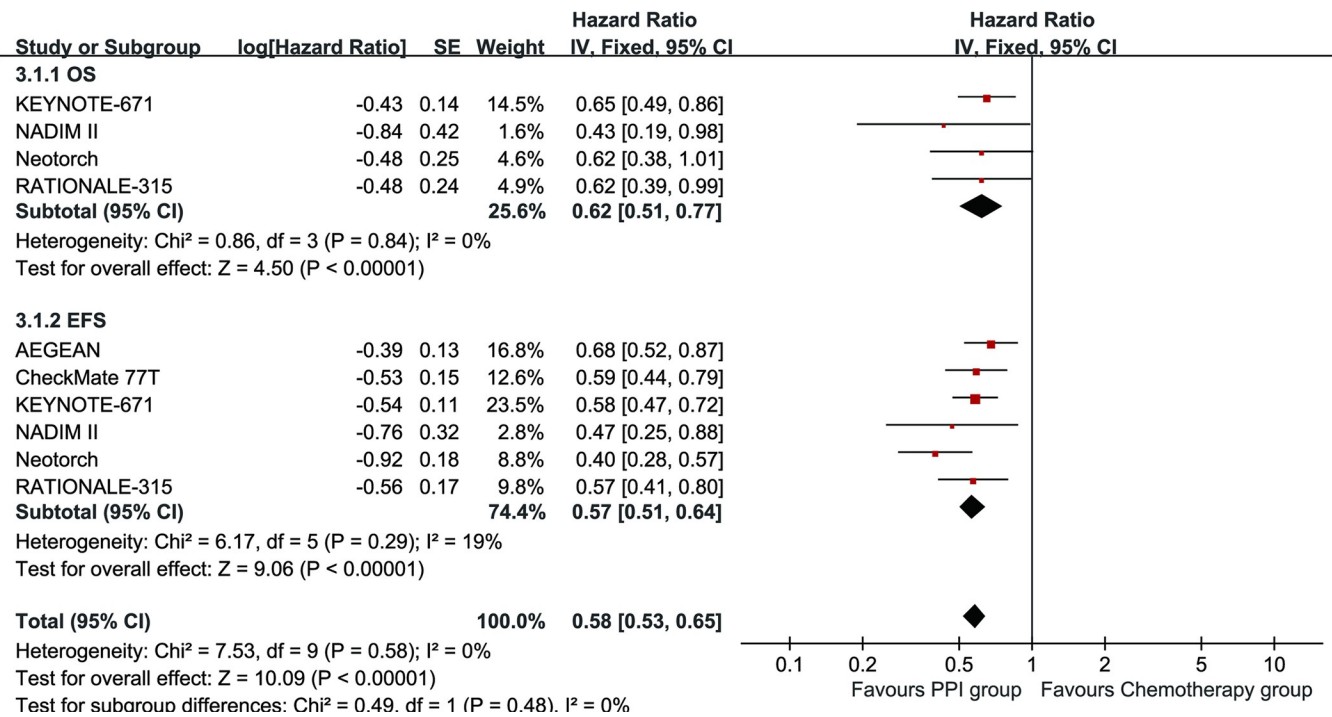

**Fig 2. Forest plots of overall survival and event-free survival associated with PPI versus chemotherapy.**

became more evident as the survival time lengthened (**Fig 3B and 3D**). The EFS advantage of the PPI group was consistent across all subgroups, particularly in the PD-L1 TPS > 50% subgroup (HR: 0.45 [0.35, 0.58]) (**Table 2**).

## Pathological responses

The ORR (RR: 2.96 [2.06, 4.26]), pathological complete response (PCR) (RR: 5.81 [4.47, 7.57]), and MPR (RR: 2.60 [1.77, 3.82]) were greater in the PPI group (**Fig 4**).

## Surgery summary

In the PPI group, there was an increase in the rates of surgery (RR: 1.05 [1.01, 1.09]) and R0 resection (RR: 1.10 [1.05, 1.15]) (**S4 Fig**).

## Safety

Overall, the PPI group had increased rates of grade 3–5 AEs (RR: 1.12 [1.04, 1.20]), serious AEs (RR: 1.34 [1.19, 1.51]), fatal AEs (RR: 1.64 [1.00, 2.68]), and discontinuations due to AEs (RR: 1.93 [1.54, 2.41]). The chemotherapy group showed a tendency for higher total AEs and dose interruptions due to AEs, but this was not statistically significant (**Table 3 and S5 Fig**).

For any grade AEs, the PPI group experienced more instances of increased AST, constipation, fatigue, cough, increased ALT, hypothyroidism, rash, pruritus, pneumonitis, hyperthyroidism, and thyroiditis (**Table 4 and S4 Table**).

For grade 3–5 AEs, the PPI group experienced more instances of pneumonitis and rash. The top 5 grade 3–5 AEs in the PPI group were decreased neutrophil count (23.15%), neutropenia (18.31%), leukopenia (6.87%), anemia (6.23%), and decreased white blood cell count (5.79%) (**Table 5 and S5 Table**).

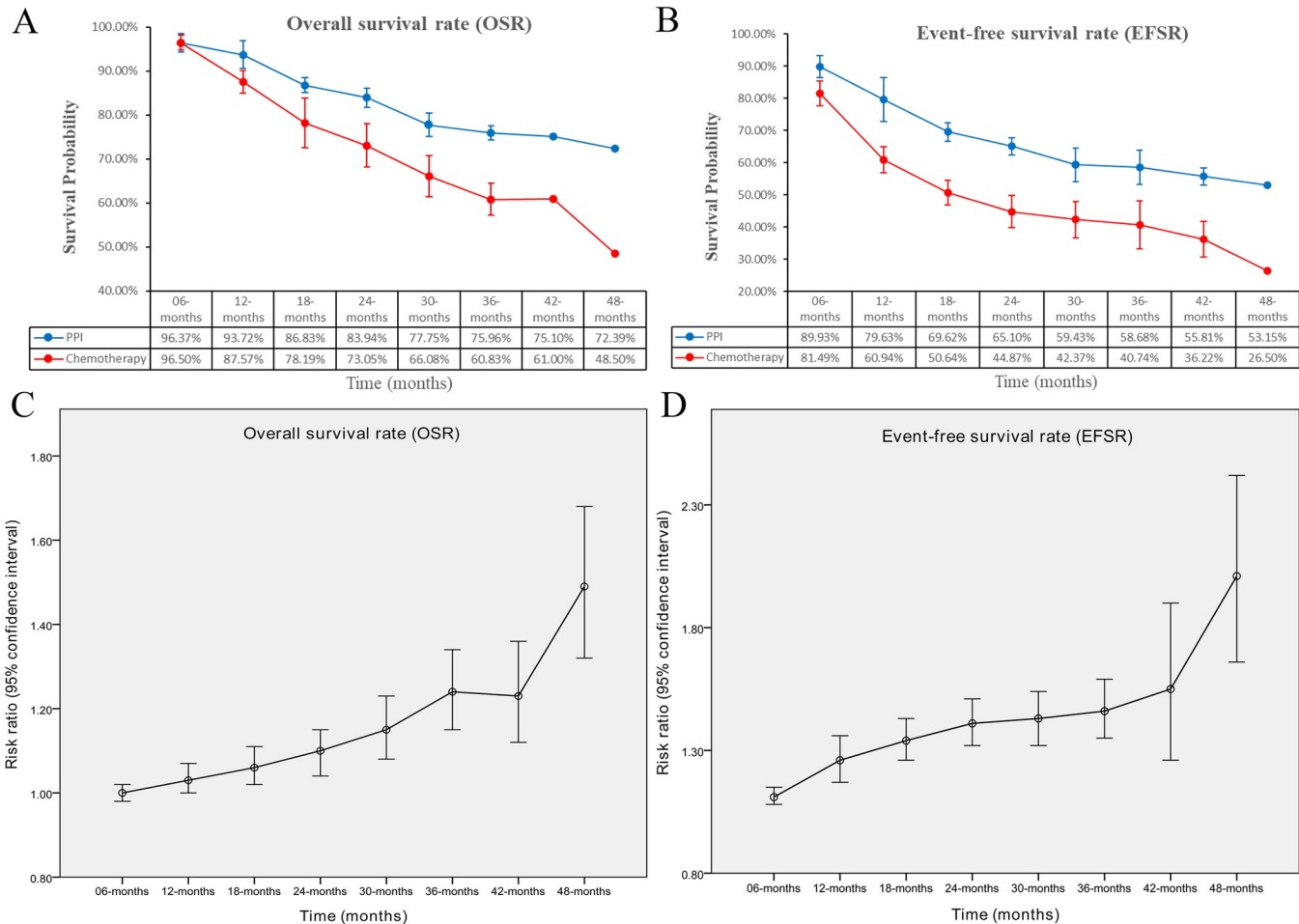

**Fig 3. Comparisons of OSR and EFSR.** (A) OSR at 6–48 months between the two groups; (B) EFSR at 6–48 months between the two groups; (C) trend of risk ratios in OSR; (D) trend of risk ratios in EFSR.

### Sensitivity analysis

Sensitivity analyses were performed for EFS (former smokers), EFSR at 12 months, and MPR. These analyses revealed that the overall reliability of the findings remained intact when any single study was excluded (**S6 Fig**).

### Publication bias

The symmetry observed in funnel plots for survival, EFSR, pathological responses, and the safety summary indicated an acceptable level of publication bias (**Fig 5**).

### Discussion

Various studies have established the efficacy of PI in both neoadjuvant and adjuvant settings [4, 5]. This finding was also corroborated by Pasqualotto et al.'s meta-analysis [6]. However, there is still a lack of large-sample evidence-based medical data for their use in the perioperative setting (neoadjuvant plus adjuvant). Our meta-analysis provides robust evidence supporting the combination of PPI with neoadjuvant chemotherapy for resectable NSCLC. The findings demonstrate significant improvements in OS and EFS, as well as enhanced

**Table 2. Subgroup analysis of event-free survival.**

| Subgroups | Event-free survival | | | | |
|---|---|---|---|---|---|
| | Included studies | Patients | HR (95% CI) | $I^2$ | P |
| **All patients** | 6 | 2941 | 0.57 [0.51, 0.65] | 18% | <0.00001 |
| **Age (year)** | | | | | |
| < 65 | 6 | 1631 | 0.55 [0.46, 0.65] | 0% | <0.00001 |
| > 65 | 6 | 1310 | 0.59 [0.49, 0.70] | 0% | <0.00001 |
| **Sex** | | | | | |
| Female | 5 | 655 | 0.64 [0.49, 0.84] | 26% | 0.001 |
| Male | 5 | 2200 | 0.56 [0.48, 0.64] | 29% | <0.00001 |
| **Smoking status** | | | | | |
| Active smoker | 4 | 535 | 0.52 [0.40, 0.70] | 0% | <0.00001 |
| Former smoker | 5 | 2000 | 0.55 [0.44, 0.69] | 54% | <0.00001 |
| Non-smoker | 6 | 406 | 0.62 [0.45, 0.87] | 43% | 0.006 |
| **ECOG PS** | | | | | |
| 0 | 4 | 1233 | 0.58 [0.48, 0.71] | 0% | <0.00001 |
| 1 | 4 | 824 | 0.56 [0.44, 0.71] | 48% | <0.00001 |
| **Race category** | | | | | |
| White | 2 | 536 | 0.53 [0.41, 0.68] | 0% | <0.00001 |
| Asian | 4 | 1508 | 0.54 [0.45, 0.65] | 27% | <0.00001 |
| Others | 3 | 1033 | 0.60 [0.49, 0.73] | 35% | <0.00001 |
| **Geographic region** | | | | | |
| Asia | 4 | 1377 | 0.51 [0.42, 0.62] | 5% | <0.00001 |
| Europe | 3 | 617 | 0.63 [0.48, 0.83] | 0% | 0.0008 |
| North America | 2 | 130 | 0.64 [0.34, 1.18] | 0% | 0.15 |
| **Pathological stage (TNM)** | | | | | |
| II | 4 | 798 | 0.66 [0.51, 0.86] | 0% | 0.002 |
| III | 6 | 2134 | 0.52 [0.45, 0.60] | 0% | <0.00001 |
| **Histologic classification** | | | | | |
| Nonsquamous | 6 | 1291 | 0.62 [0.52, 0.74] | 0% | <0.00001 |
| Squamous | 6 | 1641 | 0.53 [0.45, 0.63] | 30% | <0.00001 |
| **PD-L1 TPS** | | | | | |
| <1% | 5 | 781 | 0.75 [0.60, 0.94] | 0% | 0.01 |
| >1% | 4 | 1071 | 0.47 [0.39, 0.58] | 0% | <0.00001 |
| 1–49% | 5 | 944 | 0.52 [0.37, 0.72] | 54% | <0.00001 |
| >50% | 5 | 840 | 0.45 [0.35, 0.58] | 37% | <0.00001 |

**Abbreviations:** CI: Confidence interval; ECOG PS: Eastern Cooperative Oncology Group Performance Status; HR: Hazard ratio; PD-1: Programmed cell death protein 1; PD-L1: Programmed cell death 1 ligand 1; PPI: Perioperative PD-1/PD-L1 inhibitors; TNM: Tumor Node Metastasis; TPS: Tumor cell proportion score.

pathological responses. However, the increased rates of AEs highlight the need for careful patient selection and management strategies to mitigate potential risks.

The pooled data from six RCTs show that the PPI group significantly improves OS and EFS for resectable NSCLC. This benefit was consistent across various subgroups, particularly in patients with a PD-L1 TPS greater than 50%, suggesting that higher PD-L1 expression may predict better responses to PI. These findings are corroborated by recent studies. For example, Forde et al. demonstrated that neoadjuvant nivolumab combined with chemotherapy significantly enhanced PCR and MPR compared to chemotherapy alone, ultimately translating into

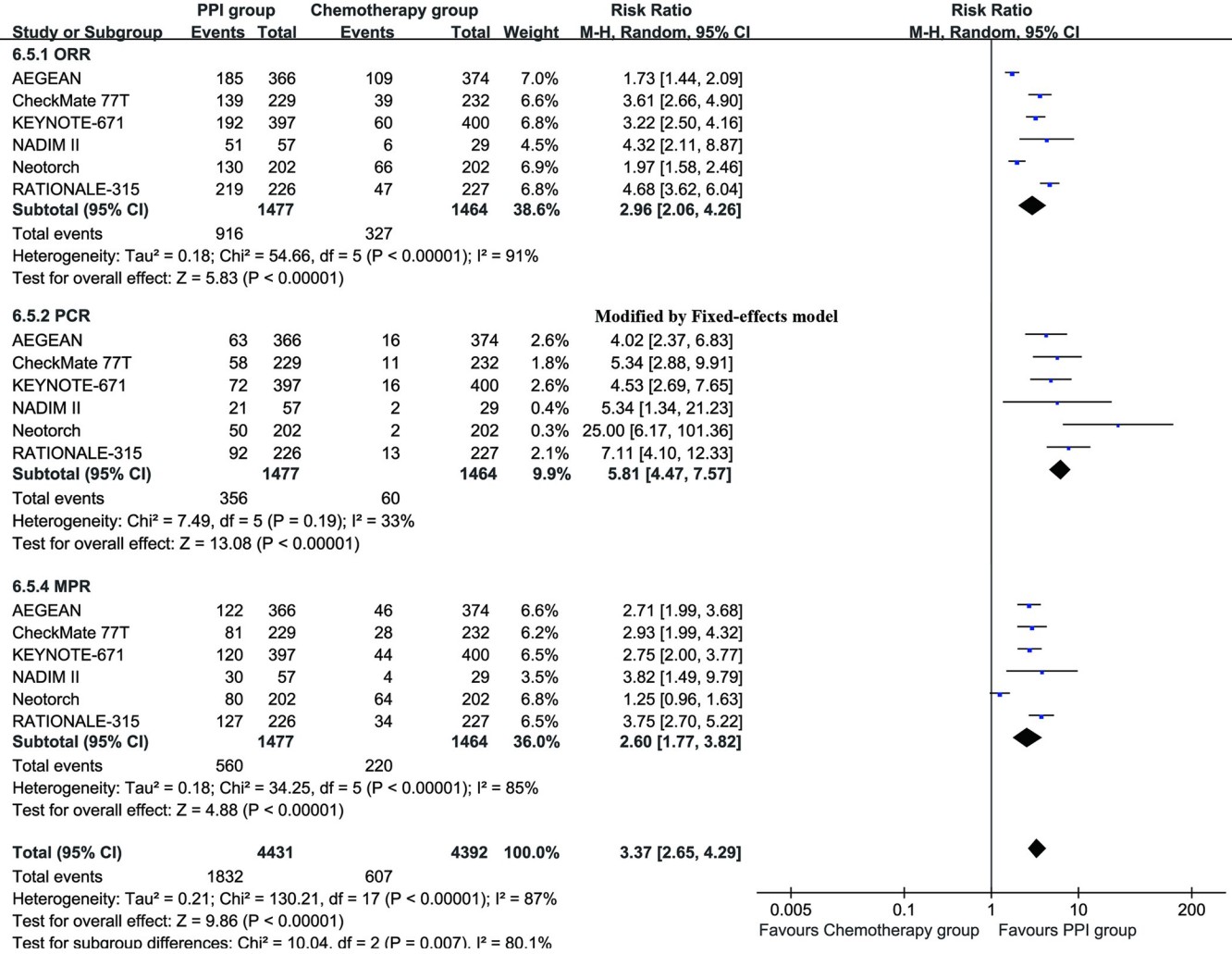

**Fig 4. Forest plots of pathological responses associated with PPI versus chemotherapy.**

**Table 3. Summary of adverse events.**

| Adverse events | PPI | | Chemotherapy | | Risk ratio [95% CI] | P |
|---|---|---|---|---|---|---|
| | Event/total | % | Event/total | % | | |
| Total adverse events | 1467/1477 | 99.32% | 1431/1464 | 97.75% | 1.01 [0.99, 1.02] | 0.31 |
| Grade 3–5 adverse events | 759/1477 | 51.39% | 682/1464 | 46.58% | 1.12 [1.04, 1.20] | 0.003 |
| Serious adverse events | 434/1420 | 30.56% | 328/1435 | 22.86% | 1.34 [1.19, 1.51] | < 0.00001 |
| Fatal adverse events | 39/1420 | 2.75% | 24/1435 | 1.67% | 1.64 [1.00, 2.68] | 0.05 |
| Discontinuation due to adverse events | 202/1420 | 14.23% | 106/1435 | 7.39% | 1.93 [1.54, 2.41] | < 0.00001 |
| Dose interruption due to adverse events | 145/428 | 33.88% | 102/429 | 23.78% | 1.50 [0.93, 2.42] | 0.09 |

**Abbreviations:** CI: confidence interval; PD-1: Programmed cell death protein 1; PD-L1: Programmed cell death 1 ligand 1; PPI: Perioperative PD-1/PD-L1 inhibitors.

**Table 4. Any grade adverse events (incidence rate > 10% in the PPI group).**

| Adverse events | PPI | | Chemotherapy | | Risk ratio [95% CI] | P |
|---|---|---|---|---|---|---|
| | Event/total | % | Event/total | % | | |
| Anemia | 565/1477 | 38.25% | 542/1464 | 37.02% | 1.05 [0.96, 1.14] | 0.34 |
| Neutrophil count decreased | 443/1218 | 36.37% | 419/1233 | 33.98% | 1.07 [0.97, 1.17] | 0.18 |
| Nausea | 481/1420 | 33.87% | 477/1435 | 33.24% | 1.02 [0.92, 1.13] | 0.72 |
| Neutropenia | 182/568 | 32.04% | 177/576 | 30.73% | 1.04 [0.88, 1.22] | 0.66 |
| AST increased | 117/428 | 27.34% | 78/429 | 18.18% | 1.50 [1.17, 1.94] | 0.002 |
| White blood cell count decreased | 312/1191 | 26.20% | 304/1203 | 25.27% | 1.03 [0.92, 1.17] | 0.6 |
| Leukopenia | 144/568 | 25.35% | 127/576 | 22.05% | 1.14 [0.95, 1.37] | 0.16 |
| Constipation | 297/1194 | 24.87% | 245/1208 | 20.28% | 1.23 [1.06, 1.42] | 0.007 |
| Alopecia | 336/1477 | 22.75% | 344/1464 | 23.50% | 0.96 [0.85, 1.09] | 0.52 |
| Fatigue | 282/1251 | 22.54% | 225/1237 | 18.19% | 1.20 [1.03, 1.40] | 0.02 |
| Arrhythmia | 58/259 | 22.39% | 53/231 | 22.94% | 1.07 [0.78, 1.48] | 0.66 |
| Decreased appetite | 255/1191 | 21.41% | 232/1203 | 19.29% | 1.14 [0.85, 1.52] | 0.4 |
| Peripheral sensory neuropathy | 49/259 | 18.92% | 39/231 | 16.88% | 1.06 [0.73, 1.54] | 0.76 |
| Cough | 101/599 | 16.86% | 74/602 | 12.29% | 1.37 [1.05, 1.79] | 0.02 |
| ALT increased | 207/1248 | 16.59% | 139/1232 | 11.28% | 1.49 [1.23, 1.81] | < 0.0001 |
| Vomiting | 149/965 | 15.44% | 126/976 | 12.91% | 1.19 [0.96, 1.49] | 0.11 |
| Platelet count decreased | 148/989 | 14.96% | 154/1001 | 15.38% | 0.97 [0.79, 1.19] | 0.77 |
| Thrombocytopenia | 78/568 | 13.73% | 74/576 | 12.85% | 1.06 [0.80, 1.42] | 0.68 |
| Asthenia | 93/763 | 12.19% | 109/774 | 14.08% | 0.87 [0.67, 1.12] | 0.27 |
| Procedural pain | 70/599 | 11.69% | 71/602 | 11.79% | 0.99 [0.73, 1.35] | 0.96 |
| Incision site pain | 111/965 | 11.50% | 99/976 | 10.14% | 1.13 [0.88, 1.46] | 0.33 |
| Hypothyroidism | 161/1477 | 10.90% | 28/1464 | 1.91% | 5.66 [3.83, 8.36] | < 0.00001 |
| Insomnia | 61/568 | 10.74% | 58/576 | 10.07% | 1.07 [0.76, 1.50] | 0.7 |
| Diarrhea | 133/1251 | 10.63% | 112/1237 | 9.05% | 1.28 [0.85, 1.93] | 0.24 |
| Rash | 133/1251 | 10.63% | 63/1237 | 5.09% | 2.08 [1.57, 2.77] | < 0.00001 |
| Pneumonia | 67/656 | 10.21% | 63/631 | 9.98% | 1.04 [0.76, 1.43] | 0.78 |

**Abbreviations:** ALT: Alanine Aminotransferase; AST: Aspartate Aminotransferase; CI: confidence interval; PD-1: Programmed cell death protein 1; PD-L1: Programmed cell death 1 ligand 1; PPI: Perioperative PD-1/PD-L1 inhibitors.

better survival outcomes [4]. Similarly, Provencio et al. indicated that perioperative PI significantly enhance survival rates in resectable NSCLC [18]. Furthermore, Efil et al. highlighted that the integration of immunotherapy in the perioperative setting leads to a substantial increase in OS and EFS, reinforcing the survival benefit observed in this meta-analysis [19]. The combined effect of chemotherapy and immunotherapy boosts tumor antigen presentation and fosters a stronger anti-tumor immune response [20, 21].

Our analysis also revealed that the PPI regimen significantly increased the rates of PCR and MPR. The RRs for PCR and MPR were 5.81 and 2.60, respectively, indicating that a greater proportion of patients achieved complete or near-complete eradication of their tumors. This is a critical finding, as pathological response has been associated with improved long-term outcomes in NSCLC. Pathological response serves as an important surrogate marker for survival in cancer treatment. Achieving a higher rate of PCR or MPR suggests that the combination therapy is effective in substantially reducing tumor burden, which is likely to translate into lower recurrence rates and better survival outcomes. This is supported by the NADIM trial, which showed that patients achieving PCR with neoadjuvant nivolumab and chemotherapy

**Table 5. Grade 3–5 adverse events (incidence rate > 1% in the PPI group).**

| Adverse events | PPI | | Chemotherapy | | Risk ratio [95% CI] | P |
|---|---|---|---|---|---|---|
| | Event/total | % | Event/total | % | | |
| Neutrophil count decreased | 282/1218 | 23.15% | 270/1233 | 21.90% | 1.05 [0.92, 1.20] | 0.44 |
| Neutropenia | 104/568 | 18.31% | 98/576 | 17.01% | 1.07 [0.84, 1.36] | 0.58 |
| Leukopenia | 39/568 | 6.87% | 28/576 | 0.05 | 1.27 [0.53, 3.03] | 0.59 |
| Anemia | 92/1477 | 6.23% | 87/1464 | 0.06 | 1.07 [0.80, 1.42] | 0.65 |
| White blood cell count decreased | 69/1191 | 5.79% | 67/1203 | 5.57% | 1.04 [0.75, 1.43] | 0.82 |
| Pneumonia | 33/599 | 5.51% | 29/602 | 4.82% | 1.14 [0.71, 1.84] | 0.59 |
| Thrombocytopenia | 20/568 | 3.52% | 16/576 | 0.03 | 1.26 [0.66, 2.41] | 0.48 |
| Platelet count decreased | 32/989 | 3.24% | 42/1001 | 4.20% | 0.77 [0.49, 1.21] | 0.26 |
| Pneumonitis | 16/828 | 1.93% | 5/834 | 0.60% | 3.22 [1.19, 8.74] | 0.02 |
| Hyperglycemia | 6/431 | 1.39% | 1/434 | 0.23% | 6.00 [0.73, 49.39] | 0.10 |
| Vomiting | 11/965 | 1.14% | 5/976 | 0.51% | 2.12 [0.77, 5.84] | 0.15 |
| ALT increased | 13/1248 | 1.04% | 5/1232 | 0.41% | 2.62 [0.94, 7.33] | 0.07 |

**Abbreviations:** ALT: Alanine Aminotransferase; AST: Aspartate Aminotransferase; CI: confidence interval; PD-1: Programmed cell death protein 1; PD-L1: Programmed cell death 1 ligand 1; PPI: Perioperative PD-1/PD-L1 inhibitors.

had significantly better EFS and OS than those who did not achieve PCR [18]. Recent studies have further validated these findings. For instance, the CheckMate 816 trial reported that neoadjuvant nivolumab plus chemotherapy led to a significant increase in MPR and PCR rates, compared to chemotherapy alone, which is consistent with our results [22]. Additionally, Gadgeel et al. highlighted that the enhanced pathological responses observed with the perioperative PPI approach can be attributed to the synergistic effects of chemotherapy and immunotherapy, which enhance tumor antigen presentation and T-cell activation [23].

Despite the efficacy benefits, the higher incidence of grade 3–5 AEs raises concerns about the safety of the PPI regimen. Common severe AEs included pneumonitis and rash, which require vigilant monitoring and management. These findings underscore the need for balancing the potential benefits of PPIs with their associated risks. The safety profile of PI has been well-documented, with immune-related adverse events (irAEs) being a notable concern. These irAEs result from the activation of the immune system against normal tissues, leading to a range of inflammatory conditions that can affect various organs, including the lungs, liver, skin, and endocrine glands [24]. In the context of perioperative treatment, the risk of irAEs must be carefully weighed against the potential survival benefits, especially since these events can greatly affect the patient's compliance with treatment and overall quality of life [25, 26].

In clinical practice, managing the safety concerns associated with PPIs involves several strategies. Early identification and prompt management of irAEs are crucial to minimizing their severity and preventing long-term complications. This requires regular monitoring of patients, educating them about the potential signs and symptoms of irAEs, and having a clear management plan in place that includes the use of immunosuppressive agents such as corticosteroids when necessary [27]. Furthermore, patient selection is critical to optimizing the safety and efficacy of perioperative PPIs. Identifying biomarkers that can predict response to therapy and the likelihood of developing severe irAEs can help tailor treatment to individual patients, thereby maximizing the therapeutic benefits while minimizing risks. For example, PD-L1 expression levels and other immune-related biomarkers have been investigated as potential predictors of response to PI [28, 29]. The use of these biomarkers can significantly improve the safety profile and efficacy of the treatment regimen [30, 31].

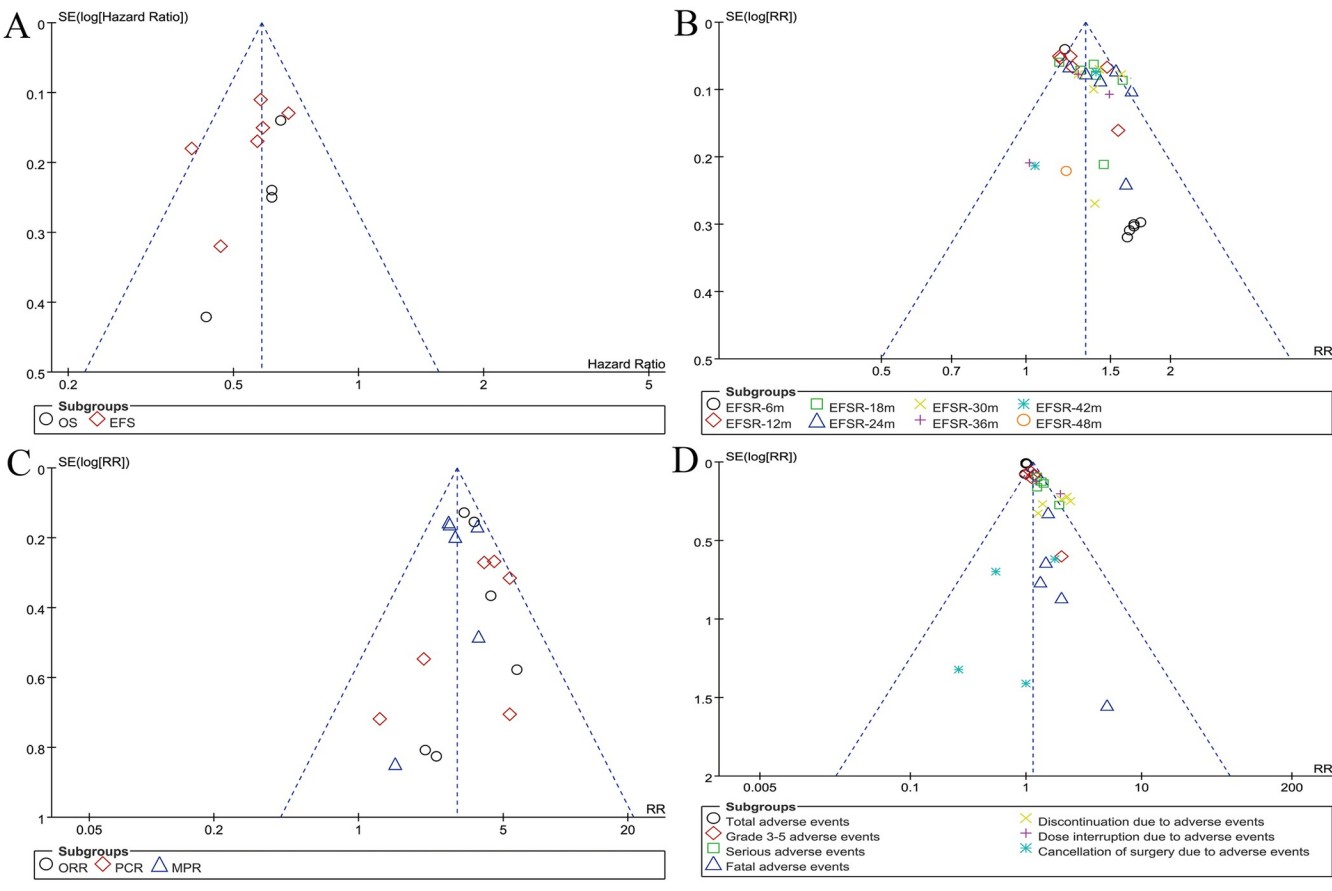

**Fig 5.** Funnel plots of survival (A), EFSR (B), pathological responses (C), and safety summary (D).

Our study has some limitations. First, restricting the review to English-language publications may have caused language bias. Second, the inclusion of RCTs that were not all phase 3 trials could influence the robustness of the outcomes. Third, the unavailability of individual patient data precluded a detailed meta-analysis, which may have limited the clinical relevance of the findings.

## Conclusion

The PPI group offers significant survival and pathological response benefits for resectable NSCLC. The survival advantages were confirmed across all subgroups and increased with longer survival times. However, the increased risk of severe AEs necessitates careful patient management and further investigation to optimize treatment protocols. Future studies should focus on improving patient selection criteria, developing methods to minimize AEs, and examining the long-term outcomes of this treatment on survival and safety.

## Supporting information

**S1 Checklist. PRISMA 2020 checklist.**
(DOCX)

**S1 Fig. Cochrane risk assessment.**
(TIF)

**S2 Fig. Forest plots of OSR at 12–48 months associated with PPI versus chemotherapy.**
(TIF)

**S3 Fig. Forest plots of EFSR at 6–48 months associated with PPI versus chemotherapy.**
(TIF)

**S4 Fig. Forest plots of surgery summary associated with PPI versus chemotherapy.**
(TIF)

**S5 Fig. Forest plots of safety summary associated with PPI versus chemotherapy.**
(TIF)

**S6 Fig.** Sensitivity analysis of EFS (Smoking status—Former smoker) (A), EFSR-12m (B), and MPR (C).
(TIF)

**S1 Table. Search strategy.**
(DOCX)

**S2 Table. Methodological quality assessments (Jadad scale) of the included studies.**
(DOC)

**S3 Table. GRADE quality assessment by therapeutic strategy and study design for the outcomes.**
(DOC)

**S4 Table. Any grade adverse events (all).**
(DOC)

**S5 Table. Grade 3–5 adverse events (all).**
(DOC)

**S1 File. Extract data details.**
(XLSX)

## Acknowledgments

The authors thank professor Wenxiong Zhang, MD (Department of Thoracic Surgery, The second affiliated hospital of Nanchang University) for his data collection and statistical advice.

## Author Contributions

**Conceptualization:** Hai Huang, Lianyun Li, Ling Tong, Houfu Luo, Huijing Luo, Qimin Zhang.

**Data curation:** Hai Huang, Lianyun Li, Ling Tong, Houfu Luo, Huijing Luo, Qimin Zhang.

**Formal analysis:** Hai Huang, Lianyun Li, Ling Tong, Houfu Luo, Huijing Luo, Qimin Zhang.

**Investigation:** Hai Huang, Qimin Zhang.

**Methodology:** Hai Huang, Qimin Zhang.

**Project administration:** Hai Huang, Qimin Zhang.

**Resources:** Hai Huang, Qimin Zhang.

**Software:** Hai Huang, Qimin Zhang.

**Supervision:** Hai Huang, Qimin Zhang.

**Validation:** Hai Huang, Qimin Zhang.

**Visualization:** Hai Huang, Qimin Zhang.

**Writing – original draft:** Hai Huang, Qimin Zhang.

**Writing – review & editing:** Hai Huang, Qimin Zhang.

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
