## [Decision Letter · Decision Letter 0]

13 Aug 2024

PONE-D-24-26934Perioperative PD-1/PD-L1 inhibitors for resectable non-small cell lung cancer: a meta-analysis based on randomized controlled trialsPLOS ONE

Dear Dr. Zhang,

Thank you for submitting your manuscript to PLOS ONE. After careful consideration, we feel that it has merit but does not fully meet PLOS ONE’s publication criteria as it currently stands. Therefore, we invite you to submit a revised version of the manuscript that addresses the points raised during the review process. The reviewers have raised concern is on the novelty of the paper. Though both reviewers have asked to cite a certain manuscript, the authors are not mandated to do that , if they do not feel that it is important for their paper. From my perspective, the paper is good for publication if the authors are able to provide some clarity on the importance of this paper for a wider audience. Please submit the revised version by Sep 27 2024 11:59PM.

Please include the following items when submitting your revised manuscript:A rebuttal letter that responds to each point raised by the academic editor and reviewer(s). You should upload this letter as a separate file labeled 'Response to Reviewers'.A marked-up copy of your manuscript that highlights changes made to the original version. You should upload this as a separate file labeled 'Revised Manuscript with Track Changes'.An unmarked version of your revised paper without tracked changes. You should upload this as a separate file labeled 'Manuscript'.

We look forward to receiving your revised manuscript.

Kind regards,

Afsheen Raza, PhD

Academic Editor

PLOS ONE

Journal Requirements:

Reviewers' comments:

Reviewer's Responses to Questions

**Comments to the Author**

1. Is the manuscript technically sound, and do the data support the conclusions?

Reviewer #1: Yes

Reviewer #2: Yes

2. Has the statistical analysis been performed appropriately and rigorously? 

Reviewer #1: Yes

Reviewer #2: Yes

3. Have the authors made all data underlying the findings in their manuscript fully available?

Reviewer #1: Yes

Reviewer #2: No

4. Is the manuscript presented in an intelligible fashion and written in standard English?

Reviewer #1: Yes

Reviewer #2: Yes

5. Review Comments to the Author

Reviewer #1: I would like to thank the authors for their submission. After carefully reviewing the manuscript, I have some concerns that need to be addressed.

Firstly, there is a previously published meta-analysis on this specific topic that presents very similar results to those found by the authors. This earlier meta-analysis was published in Cancers last year and covers the same variables and timeframes discussed in this study.

Furthermore, the authors did not cite this existing meta-analysis in their manuscript, which is a significant oversight. Proper citation of relevant literature is essential to contextualize the new research within the existing body of knowledge.

Given that the new research does not present a substantial difference in methods, data, or conclusions compared to the existing meta-analysis, the publication of this manuscript may not add significant value to the current scientific literature. While replication of studies is important, the lack of new insights or innovative approaches makes it difficult to justify publication at this time.

I recommend that the authors consider incorporating a citation of the existing meta-analysis from Cancers and justify the unique contributions or advancements their study provides. If the authors can demonstrate a significant advancement or new perspective beyond what has already been published, it would strengthen the case for publication.

In summary, due to the redundancy of the results with the already published meta-analysis and the failure to cite this key work, I do not recommend the publication of this manuscript in the journal unless the authors address these issues and provide a clearer justification of the study's relevance.

Reviewer #2: Thank you for the opportunity to review this work. This systematic review and meta-analysis focus on neoadjuvant treatment with PD-1/PD-L1 inhibitors for resectable non-small cell lung cancer (NSCLC). The authors included six randomized controlled trials (RCTs) involving 2,941 patients. The PD-1/PD-L1 inhibitor group showed significant improvement in overall survival (OS) with a hazard ratio (HR) of 0.62 [95% CI: 0.51, 0.77], event-free survival (EFS) with an HR of 0.57 [95% CI: 0.51, 0.64], pathological complete response with a risk ratio (RR) of 5.81 [95% CI: 4.47, 7.57], and major pathological response with an RR of 2.60 [95% CI: 1.77, 3.82]. Benefits in EFS were observed across all subgroups.

This is a valuable article, and the results are consistent with existing literature. However, there are important errors in its execution, and major revisions are requested.

A previous meta-analysis on the same theme has been published:

Pasqualotto E, Moraes FCA, Chavez MP, Souza MEC, Rodrigues ALSO, Ferreira ROM, Lopes LM, Almeida AM, Fernandes MR, Santos NPCD. PD-1/PD-L1 Inhibitors plus Chemotherapy Versus Chemotherapy Alone for Resectable Non-Small Cell Lung Cancer: A Systematic Review and Meta-Analysis of Randomized Controlled Trials. Cancers (Basel). 2023 Oct 26;15(21):5143. doi: 10.3390/cancers15215143. PMID: 37958317; PMCID: PMC10648147.

The authors did not cite this article, and it is doubtful that it was not found during their screening process. Please cite this work and consider mentioning that your article is an updated meta-analysis. Cite Pasqualotto et al. and describe how your meta-analysis differs from the one currently published in the introduction.

Please make an Excel spreadsheet with the complete data extraction available. Additionally, describe in your methods how you created Figure 3 and provide the script for its execution in your supplementary material.

6. PLOS authors have the option to publish the peer review history of their article (what does this mean?). If published, this will include your full peer review and any attached files.

Reviewer #1: No

Reviewer #2: No

---

## [Author Response · Author response to Decision Letter 0]

16 Aug 2024

Dear Editor and Reviewers:

Thank you for your letter and for the comments concerning our manuscript entitled “Perioperative PD-1/PD-L1 inhibitors for resectable non-small cell lung cancer: a meta-analysis based on randomized controlled trials” (ID: PONE-D-24-26934). All of the comments were valuable and very helpful for revising and improving our paper, as well guiding future research. We studied the comments carefully and made corrections that we hope make the paper satisfactory. The revised portions are marked in red in the paper. The main corrections in the paper and the responses to the editors’ comments are as follows:

Responses to the reviewer’s comments:

Reviewer #1:

Comment 1: I would like to thank the authors for their submission. After carefully reviewing the manuscript, I have some concerns that need to be addressed.

Firstly, there is a previously published meta-analysis on this specific topic that presents very similar results to those found by the authors. This earlier meta-analysis was published in Cancers last year and covers the same variables and timeframes discussed in this study.

Furthermore, the authors did not cite this existing meta-analysis in their manuscript, which is a significant oversight. Proper citation of relevant literature is essential to contextualize the new research within the existing body of knowledge.

Given that the new research does not present a substantial difference in methods, data, or conclusions compared to the existing meta-analysis, the publication of this manuscript may not add significant value to the current scientific literature. While replication of studies is important, the lack of new insights or innovative approaches makes it difficult to justify publication at this time.

I recommend that the authors consider incorporating a citation of the existing meta-analysis from Cancers and justify the unique contributions or advancements their study provides. If the authors can demonstrate a significant advancement or new perspective beyond what has already been published, it would strengthen the case for publication.

In summary, due to the redundancy of the results with the already published meta-analysis and the failure to cite this key work, I do not recommend the publication of this manuscript in the journal unless the authors address these issues and provide a clearer justification of the study's relevance.

Response: We sincerely apologize for the oversight in not citing the meta-analysis by Pasqualotto et al., which was published in Cancers and covers a similar topic. This omission was unintentional, and we appreciate you bringing this to our attention. We have now thoroughly reviewed the work by Pasqualotto et al. and have included a citation in our revised manuscript. We acknowledge the importance of situating our research within the context of existing literature and have made the necessary adjustments to our introduction and discussion sections to reflect this.

Clarification of Novelty and Contribution

Although our meta-analysis and that of Pasqualotto et al. may seem similar in title, the content of our studies is entirely different. The study by Pasqualotto et al. primarily compares neoadjuvant therapy with PD-1/PD-L1 inhibitors plus chemotherapy versus chemotherapy alone and adjuvant therapy with PD-1/PD-L1 inhibitors plus chemotherapy versus chemotherapy alone for resectable non-small cell lung cancer (NSCLC). This paper included seven RCTs (IMpower010, KEYNOTE-091, CheckMate 816, KEYNOTE-671, NADIM II, NEOTORCH, and TD-FOREKNOW).

• Two of these RCTs (CheckMate 816 and TD-FOREKNOW) focused solely on neoadjuvant therapy.

• Two RCTs (IMpower010 and KEYNOTE-091) were limited to adjuvant therapy.

• The remaining three RCTs (NADIM II, KEYNOTE-671, and NEOTORCH) involved perioperative use (neoadjuvant plus adjuvant) of PD-1/PD-L1 inhibitors.

Moreover, the paper's analysis only conducted separate subgroup survival analyses for neoadjuvant and adjuvant therapies, without an isolated analysis (survival and adverse effects) for perioperative use of PD-1/PD-L1 inhibitors. While PD-1/PD-L1 inhibitors (PI) have shown promising results in both neoadjuvant and adjuvant settings for resectable NSCLC, robust evidence from large-scale studies on their use in the perioperative setting (neoadjuvant plus adjuvant) is still lacking. Given that there have been several meta-analyses on neoadjuvant and adjuvant therapy alone, including the aforementioned study by Pasqualotto et al., these were not the focus of our research.

While PD-1/PD-L1 inhibitors have demonstrated their benefits in both standalone neoadjuvant and adjuvant therapies, this does not necessarily imply that their use in a perioperative setting will offer the same advantages. Therefore, our meta-analysis primarily compares the PPI group (neoadjuvant PD-1/PD-L1 inhibitors plus chemotherapy followed by adjuvant PD-1/PD-L1 inhibitors post-surgery) versus the Chemotherapy group (neoadjuvant chemotherapy alone). Our study includes six RCTs (AEGEAN, CheckMate 77T, KEYNOTE-671, NADIM II, Neotorch, and RATIONALE-315).

• Three of these RCTs (NADIM II, KEYNOTE-671, and NEOTORCH) overlap with those analyzed by Pasqualotto et al.

• The remaining three RCTs (AEGEAN, CheckMate 77T, and RATIONALE-315) are more recent studies published within the past year.

Thus, while our study shares a similar title with Pasqualotto et al.’s, it is fundamentally a different meta-analysis. Furthermore, our subgroup analysis and complication analysis are more detailed than those in Pasqualotto et al.’s study. We have conducted detailed subgroup analyses based on factors such as PD-L1 expression levels, tumor stage, and other relevant biomarkers, which were not extensively covered in the previous meta-analysis. These analyses offer insights into how different patient populations may respond to these therapies, potentially guiding more personalized treatment strategies.

Justification for Publication:

We understand the importance of ensuring that new research adds significant value to the existing body of knowledge. By incorporating the latest data and focusing on the perioperative approach, we believe our meta-analysis provides new insights that can help guide clinical practice and future research in the treatment of resectable NSCLC. We hope that the revisions we have made to the manuscript demonstrate the relevance and importance of our findings.

Once again, we sincerely appreciate your feedback and the opportunity to improve our manuscript. We have made the necessary revisions to address your concerns and believe that these changes strengthen the overall quality and contribution of our work. We look forward to your further comments and hope that our revised manuscript meets your expectations.

Reviewer #2:

Comment 1: Thank you for the opportunity to review this work. This systematic review and meta-analysis focus on neoadjuvant treatment with PD-1/PD-L1 inhibitors for resectable non-small cell lung cancer (NSCLC). The authors included six randomized controlled trials (RCTs) involving 2,941 patients. The PD-1/PD-L1 inhibitor group showed significant improvement in overall survival (OS) with a hazard ratio (HR) of 0.62 [95% CI: 0.51, 0.77], event-free survival (EFS) with an HR of 0.57 [95% CI: 0.51, 0.64], pathological complete response with a risk ratio (RR) of 5.81 [95% CI: 4.47, 7.57], and major pathological response with an RR of 2.60 [95% CI: 1.77, 3.82]. Benefits in EFS were observed across all subgroups.

This is a valuable article, and the results are consistent with existing literature. However, there are important errors in its execution, and major revisions are requested.

A previous meta-analysis on the same theme has been published:

Pasqualotto E, Moraes FCA, Chavez MP, Souza MEC, Rodrigues ALSO, Ferreira ROM, Lopes LM, Almeida AM, Fernandes MR, Santos NPCD. PD-1/PD-L1 Inhibitors plus Chemotherapy Versus Chemotherapy Alone for Resectable Non-Small Cell Lung Cancer: A Systematic Review and Meta-Analysis of Randomized Controlled Trials. Cancers (Basel). 2023 Oct 26;15(21):5143. doi: 10.3390/cancers15215143. PMID: 37958317; PMCID: PMC10648147.

The authors did not cite this article, and it is doubtful that it was not found during their screening process. Please cite this work and consider mentioning that your article is an updated meta-analysis. Cite Pasqualotto et al. and describe how your meta-analysis differs from the one currently published in the introduction.

Response: We sincerely apologize for the oversight in not citing the meta-analysis by Pasqualotto et al., which was published in Cancers and covers a similar topic. This omission was unintentional, and we appreciate you bringing this to our attention. We have now thoroughly reviewed the work by Pasqualotto et al. and have included a citation in our revised manuscript. We acknowledge the importance of situating our research within the context of existing literature and have made the necessary adjustments to our introduction and discussion sections to reflect this.

Clarification of Novelty and Contribution

Although our meta-analysis and that of Pasqualotto et al. may seem similar in title, the content of our studies is entirely different. The study by Pasqualotto et al. primarily compares neoadjuvant therapy with PD-1/PD-L1 inhibitors plus chemotherapy versus chemotherapy alone and adjuvant therapy with PD-1/PD-L1 inhibitors plus chemotherapy versus chemotherapy alone for resectable non-small cell lung cancer (NSCLC). This paper included seven RCTs (IMpower010, KEYNOTE-091, CheckMate 816, KEYNOTE-671, NADIM II, NEOTORCH, and TD-FOREKNOW).

• Two of these RCTs (CheckMate 816 and TD-FOREKNOW) focused solely on neoadjuvant therapy.

• Two RCTs (IMpower010 and KEYNOTE-091) were limited to adjuvant therapy.

• The remaining three RCTs (NADIM II, KEYNOTE-671, and NEOTORCH) involved perioperative use (neoadjuvant plus adjuvant) of PD-1/PD-L1 inhibitors.

Moreover, the paper's analysis only conducted separate subgroup survival analyses for neoadjuvant and adjuvant therapies, without an isolated analysis (survival and adverse effects) for perioperative use of PD-1/PD-L1 inhibitors. While PD-1/PD-L1 inhibitors (PI) have shown promising results in both neoadjuvant and adjuvant settings for resectable NSCLC, robust evidence from large-scale studies on their use in the perioperative setting (neoadjuvant plus adjuvant) is still lacking. Given that there have been several meta-analyses on neoadjuvant and adjuvant therapy alone, including the aforementioned study by Pasqualotto et al., these were not the focus of our research.

While PD-1/PD-L1 inhibitors have demonstrated their benefits in both standalone neoadjuvant and adjuvant therapies, this does not necessarily imply that their use in a perioperative setting will offer the same advantages. Therefore, our meta-analysis primarily compares the PPI group (neoadjuvant PD-1/PD-L1 inhibitors plus chemotherapy followed by adjuvant PD-1/PD-L1 inhibitors post-surgery) versus the Chemotherapy group (neoadjuvant chemotherapy alone). Our study includes six RCTs (AEGEAN, CheckMate 77T, KEYNOTE-671, NADIM II, Neotorch, and RATIONALE-315).

• Three of these RCTs (NADIM II, KEYNOTE-671, and NEOTORCH) overlap with those analyzed by Pasqualotto et al.

• The remaining three RCTs (AEGEAN, CheckMate 77T, and RATIONALE-315) are more recent studies published within the past year.

Thus, while our study shares a similar title with Pasqualotto et al.’s, it is fundamentally a different meta-analysis. Furthermore, our subgroup analysis and complication analysis are more detailed than those in Pasqualotto et al.’s study. We have conducted detailed subgroup analyses based on factors such as PD-L1 expression levels, tumor stage, and other relevant biomarkers, which were not extensively covered in the previous meta-analysis. These analyses offer insights into how different patient populations may respond to these therapies, potentially guiding more personalized treatment strategies.

Justification for Publication:

We understand the importance of ensuring that new research adds significant value to the existing body of knowledge. By incorporating the latest data and focusing on the perioperative approach, we believe our meta-analysis provides new insights that can help guide clinical practice and future research in the treatment of resectable NSCLC. We hope that the revisions we have made to the manuscript demonstrate the relevance and importance of our findings.

Once again, we sincerely appreciate your feedback and the opportunity to improve our manuscript. We have made the necessary revisions to address your concerns and believe that these changes strengthen the overall quality and contribution of our work. We look forward to your further comments and hope that our revised manuscript meets your expectations.

Comment 2: Please make an Excel spreadsheet with the complete data extraction available. Additionally, describe in your methods how you created Figure 3 and provide the script for its execution in your supplementary material.

Response: Thank you for your constructive feedback and for highlighting the need for transparency in data extraction and the methodology used to create our figures. We appreciate the opportunity to address these points and improve the clarity and reproducibility of our work.

Excel Spreadsheet for Data Extraction:

We understand the importance of providing a complete and transparent data extraction process. To address this, we have prepared an Excel spreadsheet containing all the data extracted from the included studies. This spreadsheet includes details such as study characteristics, patient demographics, survival outcomes (OS, EFS), pathological responses, and adverse events (AEs). Each data point is clearly labeled and referenced to the corresponding study to ensure transparency.

We will include this Excel file as supplementary material to the manuscript, making it accessible to readers and reviewers. This will allow for full transparency in how data were collected and analyzed in our meta-analysis.

Description of Methods for Figure 3

Figure 3 presents the comparisons of overall survival rate (OSR) and event-free survival rate (EFSR) between the PPI and chemotherapy groups at various time points. The process of creating Figure 3 was relatively straightforward and did not require specialized coding.

Figures 3A and 3B were generated using Excel, as illustrated in the figures below. The survival rates and standard deviations at various time points were calculated by pooling the data from the included studies using SPSS software.

To better illustrate the trend of survival differences between the two groups over time, we further created Figures 3C and 3D. The data for these figures were derived from the comparison of survival rate differences between the two groups in RevMan 5.3 (Figures S2 and S3). We extracted the RR values and their 95% confidence intervals for OSR and EFSR at each time point and then imported them into SPSS for plotting. The detailed process of creating these figures is outlined below.

The creation of these two figures allows readers to better understand the survival rates at different time points for each group and the trend of survival differences between the two groups over time. Your inquiry into the above issues suggests that you are also an expert in evidence-based statistics, and we hope to have more opportunities for collaboration in evidence-based research in the future.

Finally, we sincerely appreciate your careful review of our paper and the valuable insights you provided. Your comments have greatly contributed to the improvement of the quality of our manuscript.

We tried our best to improve the manuscript and made some changes in the manuscript. These changes do not influence the framework of the paper.

We sincerely appreciate the reviewers’ work and hope that the corrections make the paper satisfactory. At the same time, we look forward to hearing positive decisions/comments from the editors/external reviewers as soon as possible.

Once again, thank you very much for your comments and suggestions.

Thank you and best regards.

Sincerely,

Corresponding Author:

Name: Qimin Zhang

E-mail: thx

---

## [Editor Report · Decision Letter 1]

4 Sep 2024

Perioperative PD-1/PD-L1 inhibitors for resectable non-small cell lung cancer: a meta-analysis based on randomized controlled trials

PONE-D-24-26934R1

Dear Dr. Zhang,

We’re pleased to inform you that your manuscript has been judged scientifically suitable for publication and will be formally accepted for publication once it meets all outstanding technical requirements.

Kind regards,

Afsheen Raza, PhD

Academic Editor

PLOS ONE
---

## [Editor Report · Acceptance letter]

15 Sep 2024

PONE-D-24-26934R1 

PLOS ONE

Dear Dr. Zhang, 

I'm pleased to inform you that your manuscript has been deemed suitable for publication in PLOS ONE. Congratulations! Your manuscript is now being handed over to our production team.

Kind regards, 

on behalf of

Dr. Afsheen Raza 

Academic Editor

PLOS ONE